# GRAID: Enhancing Spatial Reasoning of VLMs through High-Fidelity Data Generation

## Abstract

Vision Language Models (VLMs) achieve strong performance on many vision-language tasks but often struggle with spatial reasoning—a prerequisite for many applications. Empirically, we find that a dataset produced by a current training data generation pipeline has a 57.6% human validation rate. These rates stem from current limitations: single-image 3D reconstruction introduces cascading modeling errors and requires wide answer tolerances, while caption-based methods require hyper-detailed annotations and suffer from generative hallucinations. We present GRAID, built on the key insight that qualitative spatial relationships can be reliably determined from 2D geometric primitives alone. By operating exclusively on 2D bounding boxes from standard object detectors, GRAID avoids both 3D reconstruction errors and generative hallucinations, resulting in datasets that are of higher quality than existing tools that produce similar datasets as validated by human evaluations. We apply our framework to the BDD100k, NuImages, and Waymo datasets, generating over 8.5 million high-quality VQA pairs creating questions spanning spatial relations, counting, ranking, and size comparisons. We evaluate one of the datasets and find it achieves 91.16% human-validated accuracy—compared to 57.6% on a dataset generated by recent work. Critically, we demonstrate that when trained on GRAID data, models learn spatial reasoning concepts that generalize: models fine-tuned on 6 question types improve on over 10 held-out types, with accuracy gains of 47.5% on BDD and 37.9% on NuImages for Llama 3.2B 11B, and when trained on all questions types, achieve improvements on several existing benchmarks such as BLINK. The GRAID framework and datasets will be available publicly after the review period.

## 1 Introduction

**SpatialVLM**

Q: How far is the a clock tower with a night view from the a bus with the street?

A: the a clock tower with a night view and the a bus with the street are 61.0 centimeters apart.

**SpatialRGPT**

Q: How wide is *<mask> <depth>*?

Image ID: 1d8d9bf7444d6051

A: It is 2.66 meters.

**GRAID**

Q: Is there at least one traffic sign to the left of any truck?

A: No

Figure 1: Examples VQA pairs from the community implementation of SpatialVLM and Spatial-RGPT, showing typical errors and issues in current synthetic data generation methods, and an example from GRAID.

Vision Language Models (VLMs) have already shown promise in a wide variety of applications, such as medical diagnosis Jin et al. (2024), biology (Maruf et al., 2025), and engineering design (Picard et al., 2025). However, despite this promise, a key failure mode of VLMs is that they are poor spatial reasoners, that is, they struggle to understand how objects are located in space and the spatial relationships between them. For example, in medical image analysis, Jin et al. (2024) found that VLMs were unable to recognize that skin lesions shown at different angles were the same pathology. Similarly, in robotics, Wang et al. (2025) found that without integrating explicit spatial

relationships, VLMs were unable to produce high-level, executable robotic task plans. As a result, without spatial reasoning, VLMs cannot be reliably deployed in embodied domains such as robotics or non-embodied domains such as medical image analysis.

While many VLMs have been trained on internet-scale data, Deitke et al. (2024) found that commonly used datasets, such as COCO (Chen et al., 2015) and Localized Narratives (Pont-Tuset et al., 2020), only contain 11 and 37 words per description on average, despite averaging 7.7 objects Lin et al. (2015) and 10.8 nouns per image. In response, there have been several recent approaches focused on generating datasets to improve the spatial reasoning of VLMs. Chen et al. (2024a) proposed SpatialVLM to generate 2 billion visual question–answer (VQA) pairs in metric space, yet our human evaluation reveals that only 57.6% of questions are valid (Section 4), with errors stemming from compounded uncertainties in depth estimation, camera calibration, and scene geometry. Cheng et al. (2025) introduced SpatialRGPT, which similarly requires 3D representations but also architectural changes to the VLM. In addition, their region-based architecture requires region-based prompting, which eliminates localization as a learned skill. SpaRE (Ogezi & Shi, 2025) generates question–answer pairs using Large Language Models (LLMs) from hyper-detailed captions but is limited in scalability since it requires extensive human effort to create the captions and inherits hallucinations from the generative models.

We introduce GRAID (**G**enerating **R**easoning questions from **A**nalysis of **I**mages via **D**iscriminative Artificial Intelligence), built on the key insight that *qualitative* spatial relationships can be reliably determined through 2D geometric analysis of bounding boxes, avoiding the metric errors and generative hallucinations commonly found in existing methods. GRAID requires only images and object detection outputs—no architectural changes, no hyper-detailed captions, and no 3D reconstruction. Table 1 offers a comparison of the differences between GRAID and prior methods. Table 1 offers a comparison of the differences between GRAID and prior methods. Our human study finds that over 91.16% of GRAID generated VQA pairs are valid as compared to under 58% of a dataset generated by a current method (Section 4). Consistent with recent benchmark findings (Ogezi & Shi, 2025), our human study implicates low-fidelity training data as the cause of a model underperforming its size class. We demonstrate GRAID at scale by applying it to Berkeley Deep Drive 100k (BDD) (Yu et al., 2020), NuImages (Caesar et al., 2019), and Waymo Open Perception (Ettinger et al., 2021), implementing 22 VQA templates spanning spatial relations, counting, ranking/extrema, localization, and size/aspect, thus generating over 8.5M pairs. To reduce compute requirements at this scale, we introduce SPARQ (Sieve Predicates And Realize Questions), a lightweight interface where question templates define `predicates` and `apply`. Shared predicates (e.g., `at_least_x_classes`) allow early rejection and yield up to $1400\times$ speedups on the heaviest templates (Section 3.2, App. Table 3). *GRAID is domain-agnostic; we instantiate on driving datasets because they provide among the largest openly available, high-quality object detection annotations at scale, not due to any AV-specific assumption in the method. In addition, the 22 templates we implement are merely to demonstrate GRAID's effectiveness as a framework; they are by no means the only VQA templates possible.*

In addition to the human study, we conduct a series of quantitative experiments to demonstrate the effectiveness of GRAID's datasets. Our experiments range from showing cross GRAID dataset generalization (RQ1), to learning simple spatial primitives that combine and lead to enhanced performance on more complex problems (RQ2). Finally, we demonstrate that training on GRAID data leads to improved VQA performance over training on datasets generated by current methods (RQ3). We fine-tune and benchmark several VLM models across a variety of tasks in existing VQA benchmarks (A-OKVQA (Schwenk et al., 2022), RealWorldQA (xAi, 2024), BLINK (Fu et al., 2024), NaturalBench (Li et al., 2024a), and VSR (Liu et al., 2023)) that challenge VLMs in both indoor and outdoor scenes far beyond the driving scenes from our exemplar source datasets. Overall, GRAID tuned models consistently outperform their counterparts tuned on data from existing methods.

In summary, this paper makes the following contributions:

1. **GRAID:** a framework that uses only 2D geometry to generate qualitative spatial VQA data, avoiding errors from single-view 3D reconstruction and hallucinations from generative models.
2. **High quality dataset:** over 8.5M VQA pairs generated from three real-world datasets, with more than 91.16% human-verified validity, making it one of the largest high-quality spatial VQA resources to date (see Sec. 4).
3. **SPARQ:** a reusable predicate library and template interface that accelerates dataset generation by early rejection of infeasible candidates, yielding up to $1400\times$ speedups on the most computationally expensive templates (see Sec. 3.2, App. Table 3).

4. **Evaluation of generalization based on GRAID:** fine-tuning on GRAID data improves VLM performance on held-out question types and on non-template tasks as well as external benchmarks, outperforming models fine-tuned on existing synthetic datasets and demonstrating knowledge transfer beyond our question templates (see Sec. 5).

Table 1: Comparison of spatial reasoning data generation frameworks

| Feature | GRAID | SpatialVLM | SpatialRGPT | SpaRE |
|---|---|---|---|---|
| Can operate on images only | ✓ | ✓ | ✓ | ✗ |
| No VLM architecture changes needed | ✓ | ✓ | ✗ | ✓ |
| No lengthy captions required | ✓ | ✓ | ✓ | ✗ |
| Avoids single-view 3D reconstruction | ✓ | ✗ | ✗ | ✓ |
| Avoids LLM-based QA gen. | ✓ | ✓ | ✓ | ✗ |
| Open-source implementation by authors | ✓ | ✗ | ✓ | ✗ |

## 2 RELATED WORK AND CHALLENGES

Whether analyzing MRI anatomical scans or planning robotic navigation, spatial reasoning is a prerequisite for embodied and non-embodied VLM deployment. Recent investigations across medical imaging (Jin et al., 2024), robotics (Wang et al., 2025), and autonomous vehicles (Jiang et al., 2025) reveal a consistent pattern: VLMs leave much to be desired in spatial understanding. To better understand these failures, recent works have investigated if VLMs can understand concepts such as physical domain understanding (Li et al., 2023), geometric understanding (Kosoy et al., 2025), and object states (Newman et al., 2024). These real-world concepts have also inspired many benchmarks like solving problems in the blink of an eye (Fu et al., 2024), naturally adversarial examples (Li et al., 2025b), physical world understanding for embodied agents (Chow et al., 2025), complex multi-step spatial concepts (Zhang et al., 2025b), and even games (Tang et al., 2025a; Lyu et al., 2025). The common finding is that VLMs leave much to be desired in terms of spatial understanding and how the physical world operates.

**3D Reconstruction** Hong et al. (2023) were among the first to teach spatial reasoning to VLMs by performing 3D scene reconstruction from multiple views then using a 3D feature extractor to connect to an LLM. While such methods worked, they required architectural changes and tons of data per scene. The authors did not specify how many images per scene were required but popular methods at the time such as Nerfstudio (Tancik et al., 2023) would have required tens to a few hundred images from known camera poses per scene. Later works avoided the requirement of many images by instead constructing implicit scene graphs: predicting depth from RGB images and instance segmentation models refine masks of detected objects, to finally lift 2D images to a 3D point clouds and perform semantic grouping. However, these approaches come at the cost of compounding errors. Gu et al. (2024) introduces ConceptGraphs but are admittedly prone to LLM and VLM hallucinations in addition to missing small and thin objects which, "impacts downstream planning". Chen et al. (2024a) introduce SpatialVLM and propose a wide acceptance metric of [50%, 200%] to account for inaccuracies of their quantitative (metric-based) questions. Despite the wide acceptance threshold, our human study reveals 57.6% of the answers generated by their community implementation are wrong (Section 4). Cheng et al. (2025) avoids many of these issues by generating their dataset from labeled 3D data, however, they propose a region-based VLM which requires architectural changes and eliminates localization as a core competency of the VLM, i.e., the user must select the object of interest rather than describe it and let the VLM find it.

**Leveraging existing data** is a more popular approach in which recent works have proposed VLMs with enhanced spatial reasoning by explicitly training them on bounding boxes (Wang et al., 2023; Yang et al., 2023b; Peng et al., 2023; Rasheed et al., 2024; Zhang et al., 2025a). Additionally, some methods have trained on point data (You et al., 2023; Deitke et al., 2024) thus becoming less dependent on bounding boxes, which may encompass with unwanted objects in object-dense scenes. However, many of these approaches leverage COCO related datasets and as Deitke et al. (2024) discovered, the sparsity of words in such source datasets are too little to contain spatial reasoning data. This led to their key insight that significantly longer human annotations are required to explicitly express spatial relationships.

## 3 GRAID

GRAID (**G**enerating **R**easoning questions from **A**nalysis of **I**mages via **D**iscriminative Artificial Intelligence) is an extensible framework that generates large-scale Visual-Question-Answering (VQA) datasets. The datasets are of higher quality than existing tools that produce similar datasets because GRAID produces valid questions and correct answers far more frequently than existing methodologies as validated by human evaluations. GRAID does this by way of two components: Scene Understanding and SPARQ. We discuss each in turn.

### 3.1 SCENE UNDERSTANDING

GRAID's key insight into reducing hallucinations in both questions and answers, is to avoid performing single-image-view 3D reconstruction —the key feature in many existing works. Instead GRAID does nearly all of its analysis in the 2D image space. In particular, GRAID merely assumes the usage of object detection models which provide class names and bounding boxes of objects in an image. Modern object detection models have achieved sufficiently high accuracy on prior global challenges such as ImageNet, and are robust enough for practical deployment, with both governments and private entities deploying popular single-stage detectors like YOLO for diverse real-world applications. Furthermore, there exists several widely accepted interpretability methods such as Saliency Maps (Li & Wong, 2024; Simonyan et al., 2014), Grad-CAM (Selvaraju et al., 2019), Grad-CAM++ (Chattopadhay et al., 2018), Score-CAM (Wang et al., 2020), SuperPixels (Hartley et al., 2021) and many more. This level of widespread deployment and tools for analysis, has yet to be achieved in the other components required to single-image-view 3D reconstruction which are not limited to but include models for depth perception, pose estimation, and plane estimation.

The problem of object detection can be formally described as follows: given an input image $I \in \mathbb{R}^{H \times W \times C}$ where $H, W,$ and $C$ denote the height, width, and number of channels, object detection models predict a set of up to $N$ bounding boxes $\mathcal{B} = \{b_i\}_{i=1}^{N}$ and their corresponding class labels $\mathcal{Y} = \{y_i\}_{i=1}^{N}$.

GRAID supports several representations of bounding boxes but for convenience, we will refer to one where each bounding box, $b_i = (x_{\min}, y_{\min}, x_{\max}, y_{\max})$, where $(x_{\min}, y_{\min})$ and $(x_{\max}, y_{\max})$ correspond to the top-left and bottom-right corners of the bounding box, respectively. Within each box, the model must also assign a class label $y_i \in \{1, \ldots, C\}$ where $C$ is the total number of class labels or object categories. This is typically formulated as a probability distribution over the label space, $p(y_i|I) = \text{softmax}(z_i)$ where $z_i \in \mathbb{R}^C$ are the raw logits from the discriminative model for class scores. Observe that $C$ is a parameter of the underlying object detection datasets and models and can easily be changed by swapping models. For example, models trained on the COCO dataset (Lin et al., 2015) have $C = 80$, whereas models trained to compete ImageNet Large Scale Visual Recognition Challenge (Russakovsky et al., 2015) have $C = 1000$.

Rather than designing a general purpose object detector or assuming a single foundational model, we build GRAID to support three of the mostly widely used computer vision packages: Detectron2, MMDetection, and Ultralytics. We define a standard interface thus allowing user's to either bring in labeled data or use their own prior trained object detection models. Note that, segmentation models can also be used as they often share the same backbone as an object detection model.

### 3.2 SPARQ

Given an image and a list of detection objects, we can now construct questions and answers based on the relationships of those bounding boxes. However, for an image with many detected objects, checking spatial relationships between objects quickly becomes expensive as this is a quadratic operation which can require comparing every object to every other object. Thus to scalably generation millions of questions in under a few hours, we design SPARQ (Sieve Predicates And Realize Questions).

**Predicates** are designed to be lightweight sanity checks before performing the full realization of a question which are more computationally expensive. For example, in the base question, `RightOf`, implemented as, *"Is there at least one {object_1} to the right of any {object_2}?"*, we can immediately check to see if there are at least two different object classes before checking spatial relationships. We can also check to see if there exists at least one pair of objects whose classes are different and their bounding boxes do not intersect (i.e., their boxes' $IoU = 0$). While these two

---

**Algorithm 1:** RIGHT–OF QUESTION REALIZER

---

**Input:** Image $I$ of width $W$ and height $H$; detections $\mathcal{D}$ (each with *label* and *bounding box*)
**Output:** List of (*question*, *answer*) pairs (possibly empty)

**1. Group detections by class**
   – Build a map $\mathcal{C}$ : label $\mapsto$ list of boxes $b = (x_{\min}, y_{\min}, x_{\max}, y_{\max})$.
   – If $|\mathrm{keys}(\mathcal{C})| < 2$, return $\emptyset$.
**2. Evaluate ordered class pairs**
   – Initialize QA $\leftarrow$ [ ].
   – For each ordered pair of distinct classes $(c_1, c_2)$:
      – Set found $\leftarrow$ False.
      – For each $b_1 \in \mathcal{C}[c_1]$ and each $b_2 \in \mathcal{C}[c_2]$:
         – Let $x_{\min}^{(1)} \leftarrow$ left edge of $b_1$, and $x_{\max}^{(2)} \leftarrow$ right edge of $b_2$.
         – If $x_{\min}^{(1)} > x_{\max}^{(2)}$ (i.e., $b_1$ is strictly to the right of $b_2$):
            – Compute $\mathrm{IoU}(b_1, b_2)$. If $\mathrm{IoU}(b_1, b_2) = 0$ (non-overlapping):
               – Append ("Is there at least one $c_1$ to the right of any $c_2$?", "Yes") to QA.
               – Set found $\leftarrow$ True and break out of the inner loops.
      – If found = False, append ("Is there at least one $c_1$ to the right of any $c_2$?", "No") to QA.
**3. Return**
   – Return QA.

---

checks are simple, their savings are significant. When generating the graid-bdd100k dataset, we find that these two predicates complete, on average, in 5.17ms, while realizing the question takes 46.95ms—nine times slower. In other questions such as `LargestAppearance`, which uses just the former predicate, the savings are more pronounced: over $1407\times$. Furthermore, we find that predicates not only saving time, but they often result sufficient conditions for the questions to be realized. In `LargestAppearance`, the predicate completes on average in 0.02ms, and 78.8% of the time results in a question being realized. In the appendix, we provide a table of GRAID-BDD (without depth) dataset that reports average predicate timing, realization time, and the share of cases where predicate success implied realization success. For the other datasets, we refer the reader to each dataset's respective README file after the review period.

**Realize Questions.** Once all predicates for a base question have succeeded, we *apply* the question—that is, we attempt to realize a question-answer pair for the image and its detected objects. One algorithm to solve the previously mentioned, `RightOf` question, is to first find the left most instance of every class of object in the image. Next, for each object found, we iterate over the remaining classes of object in the image and check for the following: 1) the bounding boxes of each potential pair should be non-overlapping, and 2) they should lie on similar planes. Observe that the second condition is necessary in the process of realizing a question as we could encounter a case where we find out that the question could be ambiguous (e.g. is an object truly the right of another if they are also on different heights?). In such instances, the questions `apply` method returns an empty list. Otherwise, when we locate a potential pair, we save them as a candidate pair until we have completed all objects in the image. The full algorithm of the `RightOf` question, is provided in Algorithm 1.

As evidence of GRAID's effectiveness, we implement over 20 base questions and apply them to three source datasets to generate over 8.5M VQA pairs. We discuss the resulting data in the next section and refer the reader to Appendix A.1 for further details of these base questions including the class name, a brief description of its predicates, and a one-line explanation of the corresponding realization algorithm.

## 4 GRAID DATASETS

The autonomous vehicle (AV) domain provides an ideal testbed for evaluating GRAID due to its exceptional wealth of high-quality, comprehensively labeled datasets that naturally capture diverse real-world scenarios. We select three prominent AV datasets —Berkeley Deep Drive (BDD) 100k, NuImages, and Waymo Open Perception—that collectively offer extensive ground truth annotations

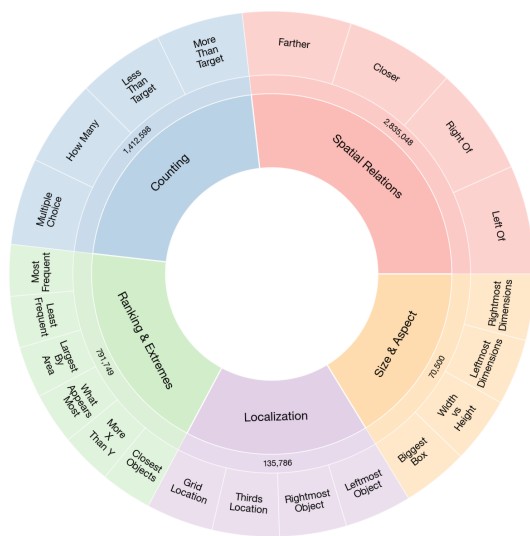

Figure 2: Hierarchical breakdown of 5.3M visual questions generated by GRAID using Berkeley Deep Drive as the source images. There are five cognitive categories: Spatial Relations (53.5%), Counting (26.7%), Ranking & Extremes (14.9%), Localization (2.6%), and Size & Aspect (1.3%). Question details including runtime, and `predicate` and `apply` methods can be found in the Appendix.

across varied driving conditions, geographical locations, and environmental factors. Additionally, the ground truth annotations in the AV space have been shown to have less human labeling errors Schubert et al. (2024) than more general datasets such as COCO. In the subsequent sections, we select to directly leverage these high-quality labels in GRAID's generation rather than train our own object detectors so that we can evaluate GRAID's effectiveness in isolation.

In total, we release six variants of GRAID generated datasets from the source datasets (see Table 2). Using BDD, we generate two variants: one without depth related questions yielding 18 classes of questions, and one with depth questions yielding 22 classes of questions. These depth questions are selected as a demonstration of GRAID's extensibility as a framework. In prior works such as SpatialVLM and SpatialRGPT, depth models are used to ask quantitative metric-based questions. Due to the inaccuracy of these models, the former proposed accepting answers that were within 50% and 200% of the estimated depth. Unfortunately, our human evaluators found that in 250 questions generated by the open implementation of SpatialVLM, over half had incorrect ground-truth answers. This is one of the main motivations for why GRAID asks qualitative rather than quantitative questions, i.e., rather than asking how far an object is in terms of metric distance, it's easier to answer which object is closer, hence the **D**iscriminative in GRAID. To further account for inaccuracies in depth models, our depth questions, like nearly all of our questions, are configurable with thresholds than can be set based on a models' confidence, a users' intuition, or domain expertise. For example, in `Closer`, we define `margin_ratio` as the configurable parameter, where the question will only be realized if the ratio of the predicted distances between the objects is at least the `margin_ratio`. This eliminates questions that appear in existing datasets which should otherwise be deemed ambiguous.

Similarly, we release the same two variants using NuImages as the source images and Waymo Open Perception. However, in Waymo, rather than using the original images, we utilize a small subset. In the Waymo Open Perception dataset, there are a few hundred unique scenes. These scenes are actually videos across six cameras on a single vehicle and so many images are repeated with just a handful of objects changing location. Thus, in our Waymo variants we select one image from the front camera with as a score that balances: (i)the number of detected objects and (ii) the ratio of the largest object area to the image area. We find this metric offers a good balance of generating more questions per image. Table 2 summarizes the various GRAID generated datasets.

HUMAN EVALUATION OF DATASET QUALITY

In order to characterize the differences between VQA datasets, we perform several kinds of human evaluations. First, we examine Huggingface to identify the most popular VQA datasets which involve spatial reasoning. At the time of submission, under the VQA dataset category, three (Li et al. (2025a); Chen et al. (2024b); Li et al. (2024b)) of the top 30 datasets ranked by downloads explicitly test for spatial reasoning. However, all three are strictly datasets and not frameworks that are capable

Table 2: GRAID Generated Datasets Overview

| Source Dataset | Question Types | # QA Pairs | Train QA | Val QA | # Train/Val Images |
|---|---|---|---|---|---|
| **BDD100k** | With Depth | **5.30M** | 4.63M | 672k | 69.9k / 9.9k |
| | Without Depth | **3.82M** | 3.34M | 485k | |
| **NuImages** | With Depth | **3.29M** | 2.65M | 641k | 60.7k / 14.9k |
| | Without Depth | **2.41M** | 1.94M | 478k | |
| **Waymo** | With Depth | **16.4k** | 13.1k | 3.33k | 798/202 |
| | Without Depth | **13.8k** | 10.9k | 2.79k | |

of generating additional data. In addition, all three utilize LLMs or VLMs in their dataset curation, leading to the question: if a VLM could already see something, is it that hard to test? A few of the remaining test for algebraic reasoning from images via tests like geometric challenges (e.g., read the sides of a triangle and use Pythagorean's theorem to solve for the missing side), however, the vast majority test for document and chart understanding, or image captioning.

In the realm of VQA generation frameworks that explicitly test for spatial reasoning from just images, we find two candidates: SpatialRGPT and SpatialVLM. There are also works such as SpaRE (Ogezi & Shi, 2025) which generate VQA questions given image and caption pairs. However observe that in Deitke et al. (2024), the authors identify that human annotations are required for better image-caption pairs, as the average word count in captions for common pairs such as COCO is merely 11 words. With such little details, methods like SpaRE leave room for LLMs to hallucinate details of an object and scene.

Our human evaluators thus evaluated the OpenSpatialDataset, the only dataset produced by SpatialRGPT, and OpenSpaces one of the more popularly used datasets generated by the community implementation of SpatialVLM. VQA examples of each dataset are shown in Figure 1. Due to the masked region queries, our evaluators were unable to ascertain the quality of the examples. In some instances, it was possible to determine if the question and answer were correct as there were only one or two regions, however, in many others, there tens of regions which often lacked semantic meaning and so identifying the subject was not possible unless a region-based prompting technique such as Set-of-Mark (Yang et al., 2023a) was used. Our evaluators were able to evaluate 50 images with 5 questions per image in OpenSpaces. An example is shown in Figure 1. The evaluators noted that most questions were not grammatically correct. Despite their best attempts to understand the question, they found $\frac{104}{250} = 41.6\%$ were not valid questions, and $\frac{144}{250} = 57.6\%$ of answers in the dataset were incorrect. Finally, of the questions that were valid, 25.2% of them had hallucinated answers. Our human evaluations corroborate recent findings from Ogezi & Shi (2025), who show that SpaceLLaVA, on average, performs the worst compared to other similarly-sized models on spatial reasoning benchmarks. Our results suggest that the poor quality of the data generated by the community implementation of SpatialVLM, which was used to train SpaceLLaVA, is a primary contributor to this performance gap.

Finally, we turn to the evaluations of a GRAID generated dataset. We ask four humans to evaluate 317 VQA pairs from the GRAID-BDD dataset without depth questions. Each person is asked for their name, which is used to compute a seed for randomly sampling the VQA pairs. As with the two previous datasets, we asked our evaluators to determine if 1) a question was valid, and 2) if the answer to the question is correct. Given that we are interested in the correctness of the question, we offer each person the person to view the image with and without bounding boxes. Without the boxes, they attempt an additional question to judge the difficulty of the questions on a Likert scale of 1 to 5. With the boxes, they can determine if the answer in the dataset is indeed correct, and if there are any labeling errors which led to a false answer. In total, our evaluators found 7 questions to be unclear, 2 questions to be invalid, and 5 labeling errors in the BDD dataset labels, i.e., over 95.58% of GRAID generated questions were valid. In terms of answers, 12 were found to be unclear and 8 were found to be invalid, hence over 93.69% of answers were valid. When we examine the unique instances (i.e., do not double count the VQA pairs with both question and answer concerns), we find that there are 28 unique instances and so in total less than 9% of the VQA pairs they evaluated were found to be either invalid or confusing. Using their feedback, we were able to address some of the ambiguities. The current public datasets have these corrections

and thus even higher validity. Lastly, our evaluators gave an average difficulty rating of 2.968, with a standard deviation of 1.146. 109 questions were marked as as a 2 or less, while 95 were marked as a 4 or higher. These results confirm that GRAID generated datasets are of the highest accuracy VQA datasets made by automated generation pipelines, and that the questions generated are of a wide variety of difficulty levels, i.e., the data avoids being uniformly easy or hard.

## 5 VISION LANGUAGE MODEL EXPERIMENTS

We conduct a series of fine-tuning experiments to determine how well a VLM can learn spatial reasoning concepts from our data. For all experiments we use Meta Llama-3.2-Vision-Instruct-11B as the base model and fine tune using LoRA Hu et al. (2021) with a learning rate of $2^{-4}$, AdamW8bit optimizer, and a linear learning rate scheduler. We ask the following research questions:

**RQ1:** Does fine-tuning on spatial reasoning tasks enable cross-dataset generalization, demonstrating acquisition of transferable spatial concepts rather than dataset-specific overfitting?

**RQ2:** Can training on fundamental spatial reasoning primitives improve performance on more complex spatial reasoning tasks not seen during training?

**RQ3:** Does training on GRAID generated datasets improve performance on established benchmarks, further validating the quality of our semi-synthetically generated training data?

**RQ1** We perform supervised fine-tuning on a limited sample of GRAID-BDD. We randomly select 10% from the training split without stratified sampling by question type. Using LoRA with rank of 16 and 200 training steps, we evaluate the model on two distinct test scenarios: (1) 1,000 held-out unstratified examples (Figure 2 provides the full distribution of questions of GRAID-BDD) from the same dataset (GRAID-BDD), and (2) 1,000 unstratified examples from an entirely different dataset (GRAID-NuImages). On the first, model performance improved dramatically from 31% to 80.7% (+49.7%), already demonstrating improved spatial reasoning capabilities. In the second, the model achieved substantial gains from 38% to 67.1% (+29.1%) on the completely unseen GRAID-NuImages dataset—which contains entirely different cities, scenes, objects, and visual contexts. These cross-dataset results strongly indicate that the model acquired transferable spatial reasoning representations rather than merely memorizing dataset-specific patterns.

**RQ2** To evaluate whether a model is truly learning spatial concepts, we select six questions to serve as our training set for supervised fine-tuning (SFT) of a Meta Llama 3.2 11B VLM: `LeftOf`, `RightOf`, `HowMany`, `AreMore`, `LargestAppearance`, and `IsObjectCentered` (full definitions are provided in Appendix A.1. We use a LoRA with rank 32, batch size 2 with 4 gradient accumulation steps, 5 warmup steps, AdamW8bit optimizer with a linear schedule, weight decay of 0.01, and train for 200 steps. Observe that these six questions yield over 18,000 VQA pairs using just GRAID-BDD (still less than half of the total training examples available), but our SFT process completes only a fraction of an epoch. At the end of our SFT, we evaluate the model on all question types in GRAID-BDD, and GRAID-NuImages, with the latter never seen in training. The results are shown in Figure 3. In nearly all questions and across both datasets, we observe wide performance increases despite only seeing six kinds of questions from only one of the datasets. These results are in agreement with findings by Tang et al. (2025b) who find that learning basic spatial concepts in simple simulated settings, leads to spatial reasoning in real world images. In both datasets, we observe a regression in `LessThanThresholdHowMany` and in GRAID-BDD, a slight regression in the same question's counterpart, `MoreThanThresholdHowMany`. Being that these two questions are some of the most common, we suspect that this is a symptom of overfitting.

**RQ3** To evaluate whether GRAID can produce datasets that transfer to real-world spatial reasoning challenges of both indoor and outdoor scenes that extend far beyond driving scenes, we supervise fine-tune (SFT) four instruction tuned VLMs, Meta Llama 3.2 11B (Grattafiori et al., 2024), Gemma 3 4B (Team et al., 2025), Qwen2.5 VL 3B (Bai et al., 2025b), and Qwen3 VL 8B (Bai et al., 2025a), on GRAID-BDD (full training details are provided in Appendix A.3). For comparison purposes, we also perform the same SFT experiment using OpenSpaces, a dataset generated by the community implementation of SpatialVLM. We evaluate all SFT variants of all models on five established VQA benchmarks which contain a variety of indoor and outdoor scenes with varying spatial reasoning complexity: BLINK (Fu et al., 2024), NaturalBench (Li et al., 2024a), A-OKVQA (Schwenk et al., 2022), RealWorldQA (xAi, 2024), and VSR (Liu et al., 2023). Rather than using GRAID's built in VLM evaluator which supports multiple prompting (zero, few-shot,

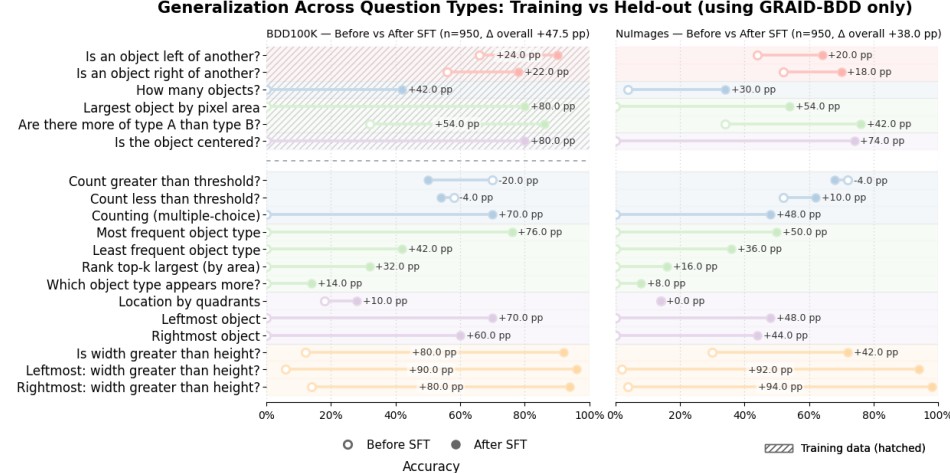

Figure 3: We fine-tune Llama 3.2 11B on only 6 questions from GRAID-BDD (hatched upper-left corner). Evaluations demonstrate a greater understanding across more difficult spatial reasoning questions in the GRAID-BDD validation set, generalization to a fifth topic not seen in training (Size & Aspect), and in all 19 question types never seen from GRAID-NuImages.

etc.) and decoding techniques (constrained, greedy, etc.), we instead follow others (Ogezi & Shi, 2025) and use VLMEvalKit (Duan et al., 2024), to ensure consistency with reported baselines. We use VLMEvalKit's exact match grading.

The results in Table 4, 5, and 6 provide further evidence that data from GRAID is of high quality as it enables substantial performance gains on VQA benchmarks across various tasks. For example, with the Llama model, we observe a significant 32.5% improvement on A-OKVQA and 15.94% overall improvement on BLINK, with particularly impressive gains on core spatial reasoning tasks: +41.13% on Relative Depth, +31.98% on Visual Correspondence, and +30.77% on Spatial Relations. We also see significant gains with the Gemma and older Qwen models, and lesser gains in Qwen 3. Despite our training data containing mostly cars and only 10 of 143 BLINK Spatial Relations questions contain the word car, the results demonstrate that GRAID generated training data captures transferrable spatial reasoning concepts rather than dataset-specific nuances. The strong improvements across all benchmarks further support that the spatial reasoning primitives learned are not specific to driving scenes and indeed apply to all kinds of scenes both indoor and outdoor. Moreover, across all four backbones, models fine-tuned on GRAID data consistently outperform those fine-tuned on the SpatialVLM dataset and, unlike OpenSpaces SFT, far less frequently incur large regressions on non-spatial tasks. Finally, the absence of overfitting to strictly driving concepts is further validated by stable performance on NaturalBench—a benchmark designed with completely adversarial examples.

## 6 CONCLUSION

In this work, we present GRAID, a simple framework for generating high-fidelity spatial reasoning VQA data from real images using only 2D detector outputs and qualitative geometry. By explicitly avoiding single-view 3D reconstruction and caption-driven synthesis, GRAID reduces cascading geometric errors and generative hallucinations while remaining easy to adopt with any object detector. Instantiated with 22 templates on three large image corpora, GRAID yields one of the largest high-quality spatial VQA datasets to date—more than 8.5M VQA pairs with over 91.16% human-verified validity—significantly higher than prior works. Supervised fine-tuning on GRAID validates that models learn spatial concepts that *transfer* beyond our templates and datasets, with consistent gains on public evaluations. As the community improves other kinds of models such as segmentation, gaze target and pose estimation, GRAID is already prepared to support those kinds of models with future templates following the SPARQ predicate and question template library. By open sourcing GRAID, we hope to further accelerate improvements in spatial reasoning so that higher-level concepts such as spatio-physical reasoning (Han et al., 2025) could be better researched.

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

# A APPENDIX

## A.1 QUESTIONS IMPLEMENTATION

`IsObjectCentered`
**Base Question:** "Divide the image into thirds. In which third does the {object_1} primarily appear? Respond with the letter only: A) left third, B) middle third, C) right third."
**Predicate:** Requires at least one object class to appear exactly once.
**Apply:** For each single-instance class, assigns A/B/C based on the bbox relative to thirds with a buffer; skips ambiguous or spanning cases.

`WidthVsHeight`
**Base Question:** "Is the width of the {object_1} appear to be larger than the height?"
**Predicate:** Requires at least one object class to appear exactly once.
**Apply:** For single-instance (optionally restricted) classes, compares width vs height; skips near-square within a threshold; returns Yes/No (supports an alternate reversed phrasing).

`LeftMost`
**Base Question:** "What is the leftmost object in the image?"
**Predicate:** At least one object class appears exactly once.
**Apply:** Finds the leftmost detection fully on the left half and separated from the second-leftmost by a margin; otherwise skips.

`RightMost`
**Base Question:** "What is the rightmost object in the image?"
**Predicate:** At least one object class appears exactly once.
**Apply:** Finds the rightmost detection fully on the right half and separated from the second-rightmost by a margin; otherwise skips.

`LargestAppearance`
**Base Question:** "If you were to draw a tight box around each object in the image, which type of object would have the biggest box?"
**Predicate:** Requires at least two different object classes.
**Apply:** Compares detection areas; returns the largest class only if its area exceeds the second by a margin.

`RankLargestK(k)`
**Base Question:** "Rank the {k} kinds of objects that appear the largest (by pixel area) in the image from largest to smallest. Provide your answer as a comma-separated list of object names only."
**Predicate:** Requires at least {k} different object classes.
**Apply:** Ranks classes by max single-instance area; asks only if each consecutive pair has a sufficient multiplicative gap.

`MostAppearance`
**Base Question:** "What kind of object appears the most frequently in the image?"
**Predicate:** Requires at least two different object classes.
**Apply:** Counts detections per class; returns the top class only if it exceeds the second by a margin.

`LeastAppearance`
**Base Question:** "What kind of object appears the least frequently in the image?"
**Predicate:** Requires at least two different object classes.
**Apply:** Counts detections per class; returns the least frequent class only if it is sufficiently below the second-least.

`LeftOf`
**Base Question:** "Is there at least one {object_1} to the left of any {object_2}?"
**Predicate:** Requires at least two classes and non-overlapping detections.
**Apply:** Answers Yes if any non-overlapping pair has {object_1}'s right edge strictly left of {object_2}'s left edge; otherwise No.

`RightOf`
**Base Question:** "Is there at least one {object_1} to the right of any {object_2}?"
**Predicate:** Requires at least two classes and non-overlapping detections.

**Apply:** Answers Yes if any non-overlapping pair has {object_1} strictly to the right of {object_2}; otherwise No.

```
HowMany
```
**Base Question:** "How many {object_1}(s) are there in this image?"
**Predicate:** At least one object class present.
**Apply:** Counts instances per class and returns (class, count) pairs.

```
AreMore
```
**Base Question:** "Are there more {object_1}(s) than {object_2}(s)?"
**Predicate:** Requires at least two object classes.
**Apply:** Pairwise compares counts; asks only when the larger exceeds the smaller by a margin; returns Yes/No accordingly.

```
WhichMore
```
**Base Question:** "What appears the most in this image: {object_1}s, {object_2}s, or {object_3}s?"
**Predicate:** Requires at least two object classes.
**Apply:** Evaluates all 3-class combinations and returns the winner only when it exceeds the runner-up by a margin.

```
Quadrants(N, M)
```
**Base Question:** "Divide the image into a grid of {N} rows x {M} columns. Number the cells from left to right, then top to bottom, starting with 1. In what cell does the {object_1} appear?"
**Predicate:** Requires a single-instance object detection.
**Apply:** Returns the 1-indexed cell if the bbox fits wholly inside one cell with margins (supports up to 12 cells); otherwise skips.

```
LeftMostWidthVsHeight
```
**Base Question:** "Does the leftmost object in the image appear to be wider than it is tall?"
**Predicate:** At least one object class appears exactly once.
**Apply:** Uses the leftmost single-instance fully on the left half; requires separation from the second-leftmost and no overlap; compares aspect ratio with a threshold; returns Yes/No (also supports reversed phrasing).

```
RightMostWidthVsHeight
```
**Base Question:** "Does the rightmost object in the image appear to be wider than it is tall?"
**Predicate:** At least one object class appears exactly once.
**Apply:** Uses the rightmost single-instance fully on the right half; requires separation from the second-rightmost and no overlap; compares aspect ratio with a threshold; returns Yes/No (also supports reversed phrasing).

```
MoreThanThresholdHowMany
```
**Base Question:** "Are there {target} or more {object_1}(s) in this image? Respond Yes/No."
**Predicate:** At least one object class present.
**Apply:** For each class with count $N > 0$, asks two targets (below and above $N$) to yield one Yes and one No, using a multiplicative threshold.

```
LessThanThresholdHowMany
```
**Base Question:** "Are there less than {target} {object_1}(s) in this image? Respond Yes/No."
**Predicate:** At least one object class present.
**Apply:** For each class with count $N > 0$, asks two targets (above and below $N$) to yield one Yes and one No; special-cases target 1 as a presence question.

```
MultiChoiceHowMany
```
**Base Question:** "How many {object_1}(s) are in the image? Choose one: A) {range_a}, B) {range_b}, C) {range_c}, D) Unsure / Not Visible. Respond with the letter only."
**Predicate:** At least one object class present.
**Apply:** For classes with $N \geq 4$, builds contiguous low/mid/high buckets (variance-adjusted), shuffles them across A/B/C, and returns the correct letter; D is provided as a fallback option.

```
ObjectsInRow
```
**Base Question:** "Are there any objects arranged in a row?"
**Predicate:** At least 3 detections.

**Apply:** Slides windows of 3+ centers, fits a line, and returns Yes if normalized vertical residual variance is below a threshold; otherwise No.

`ObjectsInLine`
**Base Question:** "Which objects appear to be arranged in a row? A) {option_a}, B) {option_b}, C) {option_c}, D) No clear row arrangement. Respond with the letter only."
**Predicate:** At least 3 detections.
**Apply:** Finds the best low-variance row of 3+ detections via linear regression; builds two distractors and returns the letter of the correct option.

`MostClusteredObjects`
**Base Question:** "Which group of objects appears most tightly clustered? A) {option_a}, B) {option_b}, C) {option_c}, D) No clear clusters. Respond with the letter only."
**Predicate:** Requires at least 9 detections.
**Apply:** Runs DBSCAN on centers with eps proportional to image diagonal; selects the most compact cluster, constructs distractors, and returns the correct letter.

`Closer`
**Base Question:** "Is there at least one {object_1} that appears closer to the camera than any {object_2}?"
**Predicate:** Requires at least two classes and non-overlapping detections.
**Apply:** Uses SAM masks and a monocular depth map to compare non-overlapping pairs; answers Yes if any {object_1} is estimated in front of a {object_2} by a margin; otherwise No.

`Farther`
**Base Question:** "Is there at least one {object_1} that appears farther from the camera than any {object_2}?"
**Predicate:** Requires at least two classes and non-overlapping detections.
**Apply:** As above but checks if {object_2} is in front; answers Yes when a {object_1} is farther than a {object_2} by a margin; otherwise No.

`DepthRanking(k)`
**Base Question:** "Rank the {k} kinds of objects that appear the closest to the camera in the image from closest to farthest. Provide your answer as a comma-separated list of object names only."
**Predicate:** Requires at least {k} different object classes.
**Apply:** Uses SAM masks and a depth map to estimate per-class closest depth; returns the top-$k$ order only if each consecutive pair differs by a sufficient margin.

## A.2 GRAID-BDD WITHOUT DEPTH REALIZATION STATISTICS

Table 3: Performance and Hit Rate Metrics for Different Question Types

| Question Type | is_applicable Avg (ms) | apply Avg (ms) | Predicate → QA Hit Rate | Empty cases |
|---|---|---|---|---|
| Divide the image into thirds. In which third does the {object_1} primarily appear? Respond with the letter only: A) left third, B) middle third, C) right third. | 0.03 | 1.82 | 71.7% | 11535 |
| Is the width of the {object_1} appear to be larger than the height? | 0.02 | 2.66 | 16.7% | 34017 |
| Divide the image into a grid of {N} rows x {M} columns. Number the cells from left to right, then top to bottom, starting with 1. In what cell does the {object_1} appear? | 0.02 | 12.28 | 42.5% | 93933 |
| If you were to draw a tight box around each object in the image, which type of object would have the biggest box? | 0.02 | 69.74 | 78.8% | 15593 |
| Rank the {k} kinds of objects that appear the largest (by pixel area) in the image from largest to smallest. Provide your answer as a comma-separated list of object names only. | 0.03 | 72.25 | 87.0% | 16663 |
| What kind of object appears the most frequently in the image? | 0.02 | 0.01 | 87.5% | 9182 |
| What kind of object appears the least frequently in the image? | 0.01 | 0.01 | 72.6% | 20133 |
| Is there at least one {object_1} to the left of any {object_2}? | 16.86 | 228.16 | 100.0% | 0 |
| Is there at least one {object_1} to the right of any {object_2}? | 16.09 | 206.98 | 100.0% | 0 |
| What is the leftmost object in the image? | 0.03 | 10.13 | 18.0% | 33486 |
| What is the rightmost object in the image? | 0.02 | 10.05 | 20.3% | 32526 |
| How many {object_1}(s) are there in this image? | 0.02 | 0.02 | 100.0% | 0 |
| Are there more {object_1}(s) than {object_2}(s) in this image? | 0.01 | 0.02 | 97.7% | 1708 |
| What appears the most in this image: {object_1}s, {object_2}s, or {object_3}s? | 0.01 | 0.02 | 69.5% | 22432 |
| Does the leftmost object in the image appear to be wider than it is tall? | 0.01 | 7.41 | 9.0% | 37131 |
| Does the rightmost object in the image appear to be wider than it is tall? | 0.02 | 6.61 | 6.6% | 38108 |
| Are there more than {target} {object_1}(s) in this image? Respond Yes/No. | 0.02 | 0.02 | 100.0% | 0 |
| Are there less than {target} {object_1}(s) in this image? Respond Yes/No. | 0.01 | 0.02 | 100.0% | 0 |
| How many {object_1}(s) are in the image? Choose one: A) {range_a}, B) {range_b}, C) {range_c}, D) Unsure / Not Visible. Respond with the letter only. | 0.01 | 0.15 | 94.3% | 4504 |

**Notes:**

- is_applicable checks if a question type can be applied to an image
- apply realizes the actual question-answer pairs
- Predicate → QA Hit Rate = Percentage of applicable cases that generated at least one QA pair
- Empty cases = Number of times predicates passed but apply realized no QA pairs

## A.3 RESEARCH QUESTION 3 TRAINING DETAILS

Here we discuss the details of our SFT using the GRAID-BDD dataset. We identify the rarest kind of question in GRAID-BDD, then randomly select that many questions from each question type yielding 51,546 training examples. We fit a LoRA of rank 32, with a batch size of 2 and 4 gradient accumulation steps. We train with a learning rate of $2^{-4}$ for 200 steps, with 5 warm up steps, with a linear schedule, weight decay of 0.01, and use the AdamW8bit optimizer.

Table 4: Performance comparison between the baseline model (Meta Llama 3.2 11B Vision Instruct), the same model fine-tuned on the OpenSpaces dataset produced by the community implementation of SpatialVLM, and the same model fine-tuned using only the GRAID-BDD dataset. All benchmarks are evaluated with VLMEvalKit using its exact match protocol.

| Dataset | Llama 3.2 | Llama+OpenSpaces | Llama+GRAID |
|---|---|---|---|
| **A-OKVQA** | 64.02% | 55.37% (-8.65) | 83.67% (+19.65) |
| **RealWorldQA** | 36.73% | 21.31% (-15.42) | 59.48% (+22.75) |
| **NaturalBench** | | | |
| *Q_Acc* | 48.97% | 15.21% (-33.76) | 50.29% (+1.32) |
| *I_Acc* | 52.82% | 15.79% (-37.03) | 53.36% (+0.54) |
| *Acc* | 73.40% | 49.25% (-24.15) | 74.28% (+0.88) |
| *G_Acc* | 23.42% | 3.63% (-19.79) | 25.42% (+2.00) |
| **BLINK** | | | |
| *Overall* | 25.72% | 25.46% (-0.26) | 42.13% (+16.41) |
| *Art Style* | 47.86% | 20.51% (-27.35) | 47.01% (-0.85%) |
| *Counting* | 25.00% | 13.33% (-11.67) | 52.50% (+25.50) |
| *Forensic Detection* | 25.76% | 26.52% (+0.76) | 26.51% (+0.75%) |
| *Functional Correspondence* | 3.08% | 16.92% (+13.84) | 24.61% (+21.53) |
| *IQ Test* | 6.67% | 25.33% (+18.66) | 18.00% (+11.33) |
| *Jigsaw* | 52.00% | 27.33% (-24.67) | 52.67% (+0.67) |
| *Multi-view Reasoning* | 35.34% | 18.05% (-17.29) | 44.36% (+9.02) |
| *Object Localization* | 61.48% | 25.41% (-36.07) | 63.11% (+1.63) |
| *Relative Depth* | 10.48% | 50.00% (+39.52) | 52.42% (+41.94) |
| *Relative Reflectance* | 0.75% | 24.63% (+23.88) | 31.34% (+30.59) |
| *Semantic Correspondence* | 12.23% | 23.02% (+10.79) | 35.97% (+23.74) |
| *Spatial Relation* | 36.36% | 18.88% (-17.48) | 72.02% (+35.66) |
| *Visual Correspondence* | 5.23% | 25.00% (+19.77) | 29.06% (+23.83) |
| *Visual Similarity* | 46.67% | 41.48% (-5.19) | 47.41% (+0.74) |
| **VSR-zeroshot** | | | |
| *Precision* | 57.35% | 54.44% (-2.91) | 52.50% (-4.85) |
| *Recall* | 95.55% | 21.46% (-74.09) | 98.57% (+3.02) |
| *Accuracy* | 61.13% | 41.98% (-19.15) | 53.36% (-7.77) |
| *F1* | 71.68% | 30.79% (-40.89) | 69.00% (-2.68) |

Table 5: Performance comparison between the baseline model (Gemma 3 4B IT), the same model fine-tuned on the OpenSpaces dataset produced by the community implementation of SpatialVLM, and and same model fine-tuned using only the GRAID-BDD dataset. All benchmarks are evaluated with VLMEvalKit using its exact match protocol.

| Dataset | Gemma 3 | Gemma+OpenSpaces | Gemma+GRAID |
|---|---|---|---|
| **A-OKVQA** | 1.57% | 53.01% (+51.44) | 76.07% (+74.50) |
| **RealWorldQA** | 13.33% | 34.90% (+21.57) | 49.02% (+35.69) |
| **NaturalBench** | | | |
| *Q_Acc* | 42.76% | 19.74% (-23.02) | 33.76% (-8.00) |
| *I_Acc* | 47.03% | 19.97% (-27.06) | 36.32% (-10.71) |
| *Acc* | 70.05% | 48.74% (-21.31) | 63.13% (-6.92) |
| *G_Acc* | 17.95% | 3.84% (-14.11) | 10.68% (-7.27) |
| **BLINK** | | | |
| *Overall* | 4.21% | 29.72% (+25.51) | 38.72% (+34.51) |
| *Art Style* | 35.90% | 48.72% (+12.82) | 50.43% (+14.53) |
| *Counting* | 10.00% | 14.17% (+4.17) | 29.17% (+19.17) |
| *Forensic Detection* | 13.64% | 13.64% (0.0) | 29.55% (+15.91) |
| *Functional Correspondence* | 0.00% | 15.38% (+15.38) | 17.69% (+17.69) |
| *IQ Test* | 0.67% | 14.00% (+13.33) | 21.33% (+20.66) |
| *Jigsaw* | 3.33% | 42.67% (+39.34) | 54.67% (+51.34) |
| *Multi-view Reasoning* | 0.75% | 36.09% (+35.34) | 39.10% (+38.35) |
| *Object Localization* | 0.00% | 22.13% (+22.13) | 48.36% (+48.36) |
| *Relative Depth* | 0.00% | 52.42% (+52.42) | 51.61% (+51.61) |
| *Relative Reflectance* | 0.00% | 34.33% (+34.33) | 28.36% (+28.36) |
| *Semantic Correspondence* | 0.72% | 20.14% (+19.42) | 31.65% (+30.93) |
| *Spatial Relation* | 0.00% | 48.25% (+48.25) | 58.74% (+58.74) |
| *Visual Correspondence* | 0.00% | 20.93% (+20.93) | 33.72% (+33.72) |
| *Visual Similarity* | 0.00% | 36.30% (+36.30) | 49.63% (+49.63) |
| **VSR-zeroshot** | | | |
| *Precision* | 54.74% | 55.56% (+0.82) | 54.15% (-0.49) |
| *Recall* | 93.64% | 27.82% (-65.82) | 78.86% (-14.78) |
| *Accuracy* | 56.87% | 48.85% (-8.02) | 54.75% (-2.12) |
| *F1* | 69.00% | 37.00% (-32.00) | 64.00% (-5.00) |

Table 6: Performance comparison between the baseline models, the same model fine-tuned on OpenSpaces, and the same model fine-tuned on the GRAID-BDD dataset. All benchmarks are evaluated using VLMEvalKit and its exact match protocol. Results are shown for four model families: Llama-3.2-11B-Vision-Instruct, Gemma-3-4B-IT, Qwen2.5-VL-3B-Instruct, and Qwen3-VL-8B-Instruct. Each cell four values corresponding to these model families in order.

| Dataset | Base | OpenSpaces-SFT | GRAID-SFT |
|---|---|---|---|
| **A-OKVQA** | 64.02% / 1.57% / 85.32% / 86.72% | 55.37% / 53.01% / 57.03% / 77.38% | 83.67% / 76.07% / 81.92% / 87.34% |
| **RealWorldQA** | 36.73% / 13.33% / 65.50% / 72.03% | 21.31% / 34.90% / 39.74% / 53.59% | 59.48% / 49.02% / 61.44% / 71.76% |
| **NaturalBench** | | | |
| Q_Acc | 48.97% / 42.76% / 51.39% / 61.89% | 15.21% / 19.74% / 21.34% / 39.42% | 50.29% / 33.76% / 47.45% / 58.97% |
| I_Acc | 52.82% / 47.03% / 55.23% / 63.87% | 15.79% / 19.97% / 21.74% / 41.16% | 53.36% / 36.32% / 50.37% / 61.08% |
| Acc | 73.40% / 70.05% / 74.46% / 80.09% | 49.25% / 48.74% / 52.39% / 65.76% | 74.28% / 63.13% / 71.21% / 78.50% |
| G_Acc | 23.42% / 17.95% / 25.63% / 37.37% | 3.63% / 3.84% / 5.42% / 15.74% | 25.42% / 10.68% / 23.05% / 35.05% |
| **BLINK** | | | |
| Overall | 25.72% / 4.21% / 49.18% / 56.71% | 25.46% / 29.72% / 37.30% / 42.98% | 42.14% / 38.72% / 44.45% / 62.28% |
| Art Style | 47.86% / 35.90% / 56.41% / 43.59% | 20.51% / 48.72% / 46.15% / 50.43% | 47.01% / 50.43% / 56.41% / 72.65% |
| Counting | 25.00% / 10.00% / 68.33% / 65.00% | 13.33% / 14.17% / 48.33% / 45.83% | 52.50% / 29.17% / 61.67% / 64.17% |
| Forensic Detection | 25.76% / 13.64% / 32.57% / 89.39% | 26.52% / 13.64% / 21.21% / 28.03% | 26.52% / 29.55% / 20.45% / 75.76% |
| Functional Correspondence | 3.08% / 0.00% / 23.84% / 3.08% | 16.92% / 15.38% / 18.46% / 28.46% | 24.62% / 17.69% / 29.23% / 36.15% |
| IQ Test | 6.67% / 0.67% / 26.00% / 0.00% | 25.33% / 14.00% / 27.33% / 28.00% | 18.00% / 21.33% / 18.67% / 26.67% |
| Jigsaw | 52.00% / 3.33% / 50.00% / 69.33% | 27.33% / 42.67% / 54.00% / 39.33% | 52.67% / 54.67% / 48.67% / 62.67% |
| Multi-view Reasoning | 35.34% / 0.75% / 48.12% / 54.14% | 18.05% / 36.09% / 46.62% / 45.86% | 44.36% / 39.10% / 46.62% / 50.38% |
| Object Localization | 61.48% / 0.00% / 54.91% / 68.03% | 25.41% / 22.13% / 35.25% / 59.02% | 63.11% / 48.36% / 50.00% / 67.21% |
| Relative Depth | 10.48% / 0.00% / 70.96% / 87.90% | 50.00% / 52.42% / 57.26% / 51.61% | 52.42% / 51.61% / 60.48% / 86.29% |
| Relative Reflectance | 0.75% / 0.00% / 39.55% / 32.84% | 24.63% / 34.33% / 32.84% / 38.06% | 31.34% / 28.36% / 40.30% / 33.58% |
| Semantic Correspondence | 12.23% / 0.72% / 31.65% / 17.99% | 23.02% / 20.14% / 24.46% / 29.50% | 35.97% / 31.65% / 29.50% / 47.48% |
| Spatial Relation | 36.36% / 0.00% / 83.21% / 86.01% | 18.88% / 48.25% / 48.25% / 54.55% | 72.03% / 58.74% / 75.52% / 82.52% |
| Visual Correspondence | 5.23% / 0.00% / 40.11% / 86.63% | 25.00% / 20.93% / 20.93% / 49.42% | 29.07% / 33.72% / 36.05% / 84.30% |
| Visual Similarity | 46.67% / 0.00% / 70.37% / 87.41% | 41.48% / 36.30% / 47.41% / 56.30% | 47.41% / 49.63% / 56.30% / 82.22% |
| **VSR-zeroshot** | | | |
| Precision | 57.35% / 54.74% / 78.08% / 88.58% | 54.44% / 55.56% / 55.66% / 65.44% | 52.50% / 54.15% / 68.34% / 80.35% |
| Recall | 95.55% / 93.64% / 80.44% / 85.06% | 21.46% / 27.82% / 18.76% / 31.00% | 98.57% / 78.86% / 84.42% / 88.39% |
| Accuracy | 61.13% / 56.87% / 78.31% / 86.67% | 41.98% / 48.85% / 50.33% / 56.06% | 53.36% / 54.75% / 71.85% / 82.90% |
| F1 | 71.68% / 69.09% / 79.24% / 86.78% | 30.79% / 37.08% / 28.06% / 42.07% | 69.00% / 64.21% / 75.53% / 84.18% |

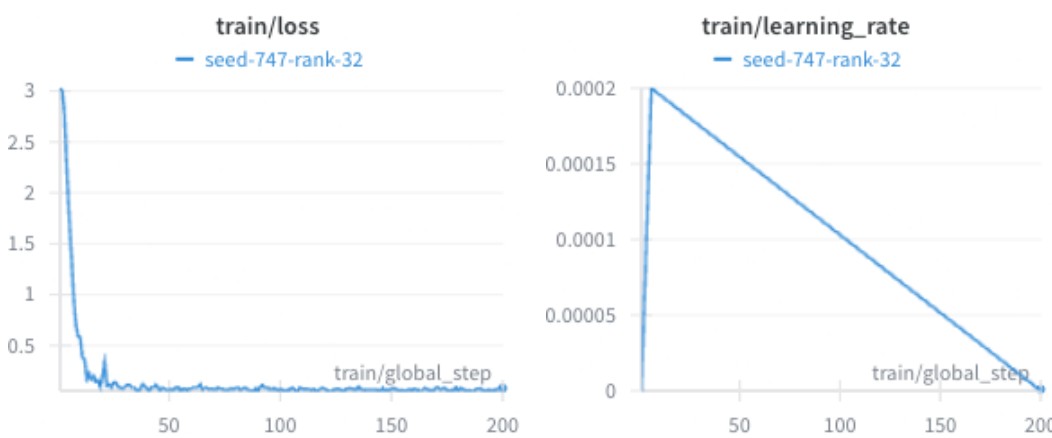

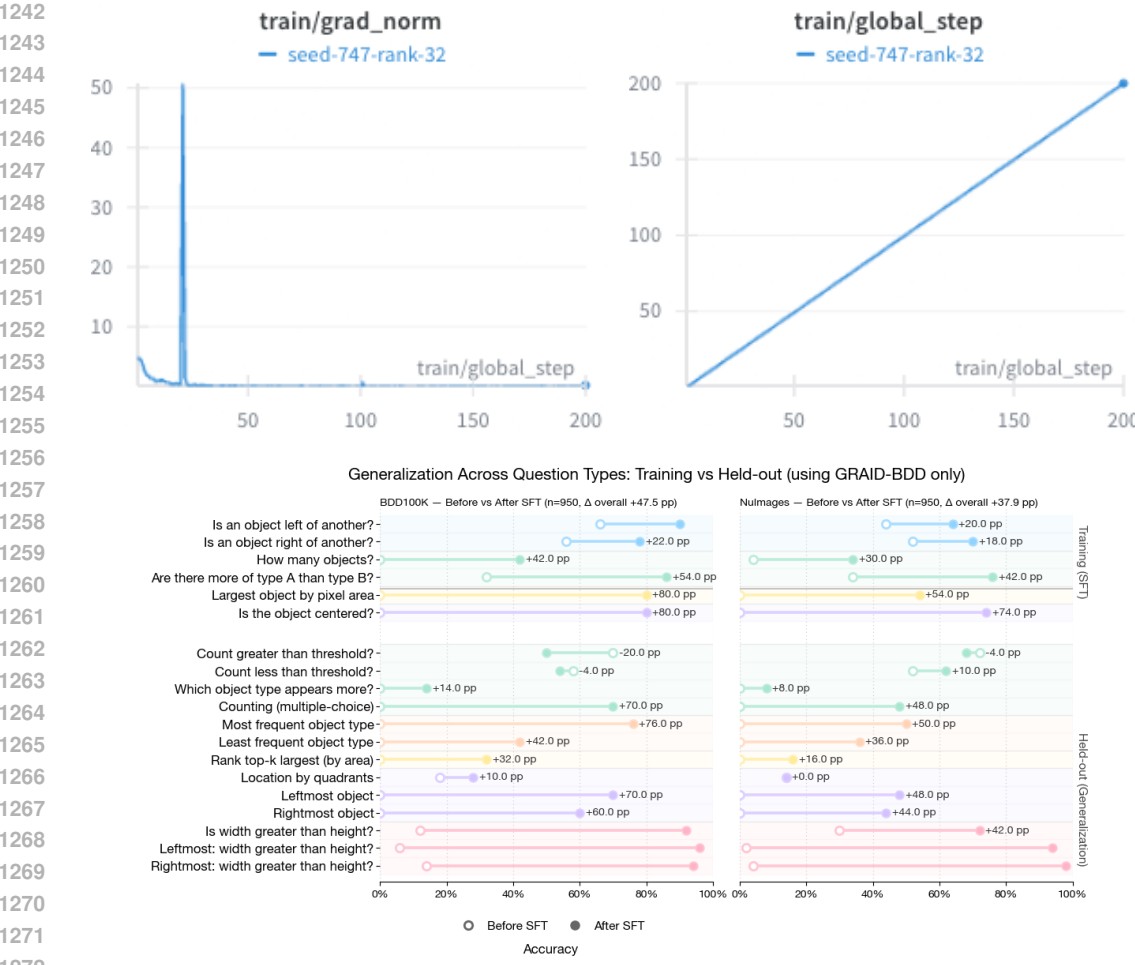

Figure 4: After supervised fine-tuning on the GRAID-BDD dataset, we can see improvements in the model's ability to answer questions on the held-out questions of GRAID-BDD, and a dataset with a different distribution of scenes, GRAID-NuImages.

## A.4 VLM TRAINING ABLATIONS

In all our supervised fine-tuning experiments, we use `unsloth` to training our LoRA adapters. In this section, we discuss ablations on the various components we can have LoRA adapters for: vision layers, language layers, attention modules, and mlp modules. In each of the experiments, we enable SFT of all components except one at a time. All experiments use a rank of 16, batch size of 2, 4 gradient accumulation steps, 5 warmup steps, 200 steps, a learning rate of $2^{-4}$, a linear scheduler, AdamW8bit optimizer, and 0.01 weight decay. In the charts below, we see that the training loss curves for all experiments except one are identical: disallowing the fine-tuning of the language layers. In this setting, we are unable to train the model as well as in the others, and the gradient norm remains relatively high. These results hint that the vision layers of a VLM can use some improvement, however, the vast majority of spatial reasoning is still occurring the language space of the model, and not its vision encoder.

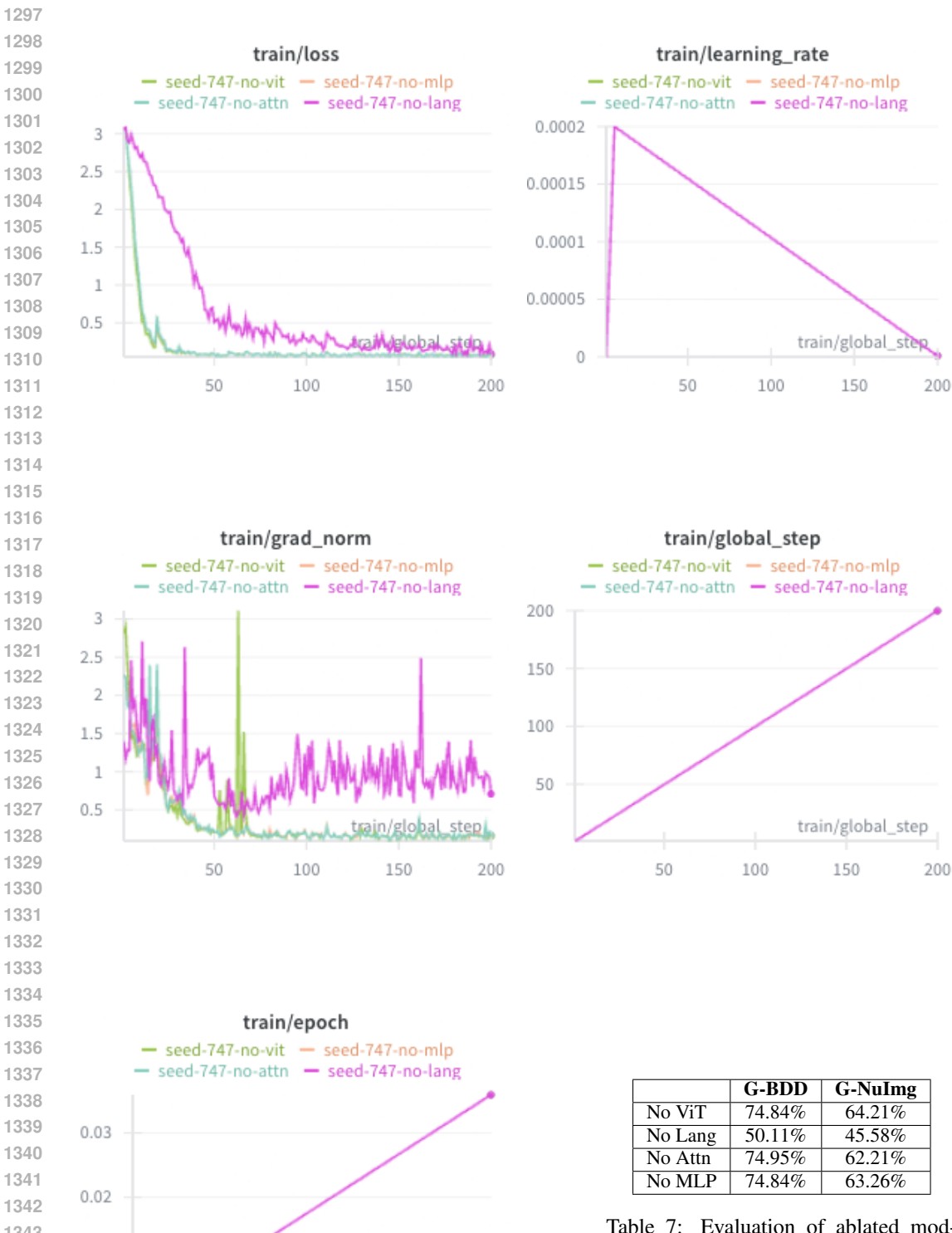

|  | G-BDD | G-NuImg |
|---|---|---|
| No ViT | 74.84% | 64.21% |
| No Lang | 50.11% | 45.58% |
| No Attn | 74.95% | 62.21% |
| No MLP | 74.84% | 63.26% |

Table 7: Evaluation of ablated models on GRAID-BDD and GRAID-NuImages datasets