# OpenReview forum: "GRAID: Enhancing Spatial Reasoning of VLMs through High-Fidelity Data Generation"
_ICLR.cc/2026/Conference — Submitted to ICLR 2026_

### Official Review · Reviewer_8c6y · 2025-10-17

**Soundness:** 2
**Presentation:** 2
**Contribution:** 2
**Rating:** 2
**Confidence:** 4

**Summary:**

The paper introduces GRAID, a framework that generates high-fidelity, qualitative spatial reasoning VQA data using only 2D bounding boxes and a predicate-based accelerator (SPARQ), avoiding errors from single-view 3D reconstruction and LLM-generated captions. Across BDD100k, NuImages, and Waymo, it produces 8.5M+ VQA pairs with ~91% human-validated accuracy and, after SFT on Llama 3.2-Vision-11B, yields strong generalization to unseen question types and external benchmarks.

**Strengths:**

- Clear and compelling motivation: the paper precisely identifies why prior pipelines underperform—single-image 3D reconstruction compounds modeling errors and forces wide tolerances, while caption-based methods demand hyper-detailed annotations and inherit generative hallucinations—and proposes a clean 2D, qualitative alternative.

- RQ1 in Section 5 is crucial and is lacking in many current approaches.

**Weaknesses:**

- Incomplete experiments
  - The benchmarks used for testing are too few.
  - More spatial understanding benchmarks are needed, especially to show that autonomous driving data can generalize to indoor scenes.
  - Only Llama-3.2-Vision-Instruct was fine-tuned, which limits the persuasiveness of the results.
  - There is no comparison against any methods specifically designed to improve VLM spatial reasoning (at the capability-improvement level).
- Presentation issues
  - The paper includes many unnecessary details in the main text that take up substantial space.
  - There are too few experimental results.
  - There is no table presenting results in the main text.
  - Some images in the appendix are blurry.

**Questions:**

- Add comparisons against methods explicitly aimed at improving VLM spatial reasoning (at the capability-improvement level).
- Expand the benchmarking suite with more spatial understanding benchmarks, including indoor scene datasets, to assess cross-domain generalization.
- Move non-essential implementation details to the appendix to improve clarity and flow.
- Fine-tune and report results on additional base models beyond Llama-3.2-Vision-Instruct to strengthen evidence.
- Provide clearer figures in the appendix.
- Report more experimental results, including ablations on predicate sets, sensitivity to box quality, and training data size.

---

> ### Author Response · Authors · 2025-11-20
>
> We thank the reviewer for their detailed feedback and for highlighting our motivation and RQ1 as strengths.
>
> ### 1. "Too few benchmarks"
>
> We agree that evaluation on strong spatial benchmarks is crucial. In the current submission, we already evaluate on four recent VQA benchmarks with substantial spatial content:
> - BLINK – a multi-task spatial reasoning benchmark containing diverse scenes, including many non-driving and indoor-like settings, with tasks such as Spatial Relation, Relative Depth, Visual/Relative Correspondence, Object Localization, IQ Test, and Jigsaw
> - RealWorldQA – real-world images focusing on fine-grained physical and spatial commonsense
> - A-OKVQA – open-ended questions over varied everyday scenes
> - NaturalBench – an adversarial benchmark designed to probe robustness and avoid superficial shortcuts
>
> As summarized in Table 3, SFT on GRAID data yields substantial gains on BLINK (especially on Relative Depth, Spatial Relations, and Visual Correspondence), RealWorldQA, and A-OKVQA while maintaining stable performance on NaturalBench. Crucially, the GRAID question templates themselves are not specific to autonomous driving: they operate on generic 2D object detections (left/right, counting, extrema, localization, etc.) and do not reference lanes, road markings, or driving-specific semantics. RQ1 shows cross-dataset generalization from GRAID-BDD to GRAID-NuImages (different cities and scenes), and RQ2 shows cross-question-type generalization from six primitive templates to over 10 held-out types. RQ3 further shows that a model fine-tuned on GRAID VQA pairs learns general spatial concepts that transfer to several non-driving, mixed-domain benchmarks.
>
> We will emphasize in the revision that (i) our spatial predicates are domain-agnostic, and (ii) the selected benchmarks already include many non-driving and indoor-style scenarios.
>
> ### 2. “No comparison against methods explicitly aimed at improving VLM spatial reasoning”
>
> We do compare against such methods at the data-generation and dataset quality level. Figure 1 qualitatively contrasts GRAID-generated questions with examples from SpatialRGPT and the community implementation of SpatialVLM, highlighting typical issues from region-based prompting and 3D reconstruction. Table 1 explicitly compares GRAID to SpatialVLM, SpatialRGPT, and SpaRE along axes such as 3D requirement, reliance on hyper-detailed captions, and whether they use generative models in the data pipeline. Section 4 presents a human study directly evaluating QA validity and answer correctness for both GRAID and data produced by a recent spatial VQA pipeline (SpatialVLM), showing that only 57.6% of their answers are correct versus over 91% valid QA pairs for GRAID-BDD.
>
> 3. “Too few experimental results” / “no table presenting results in the main text”
>
> The current version is organized around three research questions in Section 5, each backed by its own experiment:
> - RQ1: Cross-dataset generalization from GRAID-BDD to GRAID-NuImages, showing large gains after fine-tuning on only 10% of GRAID-BDD
> - RQ2: Training on 6 primitive question types and evaluating on >10 held-out spatial templates to test concept compositionality
> - RQ3: Transfer to four external benchmarks with detailed per-subtask results summarized in Table 3
>
> Contrary to the comment, the main text already includes two tables and one in the appendix:
> - Table 1: Comparison of GRAID with prior spatial VQA data-generation methods
> - Table 2: Quantitative results for RQ1/RQ2 on GRAID-BDD and GRAID-NuImages
> - Table 3: (appendix, referenced from the main text): Performance comparison between the baseline Meta Llama 3.2 11B and our GRAID-SFT model across four VQA benchmarks
>
> 4. “Only Llama 3.2 was fine-tuned”
> We agree that evaluating multiple VLM backbones strengthens the empirical case. We will extend our experiments in the revised version to an additional recent open-weight VLM. This will help confirm that the benefits of GRAID are not model-specific.
>
> 5. Presentation issues
>
> We appreciate the comments on the presentation. We will move some implementation details from the main text into the appendix. We will regenerate blurry images at higher resolutions and improve figure fonts and captions for readability. With regards to ablations (predicate sets, data size, box quality), RQ2 already acts as a predicate-set ablation, training only on 6 primitive question types and evaluating on a broader set of spatial questions; we will make this interpretation explicit. RQ1 inherently varies training data size and demonstrates strong gains even with limited data; we will clarify this as partial evidence of data-efficiency. With regards to box quality, as discussed in other reviews, observe that even using ground-truth annotations does not guarantee correctness as there are errors in the annotations.
>
> We hope these clarifications address the reviewer’s concerns and better highlight the breadth of our experiments and comparisons.

---

> > ### Comment · Reviewer_8c6y · 2025-11-21
> > **Response to the Authors**
> >
> > Thank you for your response.
> >
> > First, instead of saying that you will make modifications, I suggest directly uploading an updated PDF file with the changes.
> >
> > Second, please respond to my questions directly, rather than misrepresenting my point. My original comment was:
> >
> > > “There is no comparison against any methods specifically designed to improve VLM spatial reasoning (at the capability-improvement level).”
> >
> > I made it clear that I was referring to comparisons **at the capability-improvement level for the model**. In your response, you wrote:
> >
> > > “We do compare against such methods at the data-generation and dataset quality level.”
> >
> > I am already aware of your comparisons at the data-generation and dataset quality level. However, my main concern is that there are too few experimental results and no table presenting results in the main text regarding model testing and comparisons with other data-generation approaches at the capability-improvement level for the model.
> >
> > Comparing data quality alone is insufficient; only demonstrating significant improvements in model capability can convincingly show the value of your data

---

### Official Review · Reviewer_eQNk · 2025-10-31

**Soundness:** 3
**Presentation:** 4
**Contribution:** 3
**Rating:** 8
**Confidence:** 3

**Summary:**

This paper proposes a data generation pipeline to enhance the spatial reasoning capability of VLMs. In contrast to previous method, the proposed pipeline only relies on 2D bounding box detections (and monocular depth models in some cases), and only generate questions asking about qualitative spatial relationships. Therefore, the quality of the VQA pairs is much higher (confirmed by human studies). The authors run their data pipeline on 3 large-scale autonomous driving datasets to generate data for SFT on VLMs. Experiment results are very promising, as the finetuned model works significantly better on existing benchmarks such as BLINK.

**Strengths:**

1. The paper proposes a simple yet effective framework to generate high-quality data from only 2D bounding boxes. Although I don't find the data generation pipeline itself to be novel, it does solve the core problems of low data quality of the previous spatial VQA datasets in a simple and intuitive way.
2. The experiments conducted are very sound and support the claim. I'm especially impressed by the human studies showing the flaws of previous VQA datasets, and the 95% accuracy of the proposed dataset. It is also impressive that the model finetuned on SPARQ can generalize to datasets such as BLINK, which has a large domain gap from the autonomous driving datasets that the model is finetuned on. This shows a good potential of the proposed approach.

**Weaknesses:**

1. My main concerns of the proposed pipeline is that it is only evaluated on autonomous driving datasets. The authors claim in section 3.1 that GRAID can also work on detection-model-generated bounding boxes, but it is unclear how much the data quality will degrade when switching from GT detections to model detections. Therefore, I'm concerned about the generalization of the proposed method beyond autonomous driving scenes. One possible experiment the authors can do is: similar to L339-389, the authors can ask humans to judge the QA pair quality, but with bounding boxes from model detection.

**Questions:**

N/A

**Details Of Ethics Concerns:**

The authors do human studies to evaluate the VQA data quality of several datasets. It is possible that the image content from some of them could make the human judge uncomfortable. Therefore, ICLR should check if the authors have obtained IRB approval or other similar regulations on human studies.

---

> ### Author Response · Authors · 2025-11-20
>
> We thank the reviewer for the positive assessment and for clearly summarizing both the motivation and the empirical findings. We are especially glad that the strengths around data quality, human studies, and cross-domain generalization to BLINK came through clearly.
>
> Regarding the concern about generalization beyond autonomous driving and the use of model detections, we agree this is an important dimension. Our main experiments focus on driving datasets because they are large-scale, come with high-quality instance-level annotations, and allow us to isolate the effect of spatial reasoning from detection noise. As we also note in our response to Reviewer Yb6x, even these ground-truth annotations (like many large benchmarks, not just in detection) contain some label errors. Despite this, GRAID is still able to generate high-fidelity QA pairs that yield meaningful gains on established benchmarks that not only include spatial reasoning but also other domains like IQ tests and real world VQA, suggesting robustness to a realistic level of underlying annotation noise.
> GRAID is explicitly designed as a framework that can operate on any set of 2D detections, including those produced by off-the-shelf detectors in other domains. In such settings, the QA quality will naturally track the quality of the underlying detector: if objects are systematically missed or badly localized, some candidate questions will become invalid or ambiguous. An interesting future direction could be to study this effect in a domain-specific setting and design predicates/realizers that are robust to noisy detections given a prior over the scene distribution (e.g., indoor scenes, different camera geometries). For example, in an indoor scene, one may want to ask which object is closest to the {object_1} (e.g., coffee table). If the detector tends to miss many small objects, one could instantiate this as a multiple-choice question over only those objects whose detections exceed a chosen confidence (and size) threshold, thereby reducing the impact of noisy or missing detections.
>
> Regarding the ethics flag, our human studies are conducted on images from datasets that are already publicly available including any PII that might be in them. Our participants only judged question validity, answer correctness, and perceived difficulty. The tasks involve minimal risk and no personally identifying information. We followed our institution’s guidelines for human-subject research; we will add a short clarification of this in the ethics/broader impact section.

---

> > ### Comment · Reviewer_eQNk · 2025-11-25
> >
> > Thank the authors for their efforts in the rebuttal. After reading the response and the reviews from other reviewers, I find my original concerns about the limited scope of evaluation (i.e., only on autonomous driving datasets with ground-truth bbox) are not fully addressed. While the authors provide explanations that the proposed method could work on other domains, there are no direct experimental results or evidence to directly support that.

---

### Official Review · Reviewer_kpWh · 2025-11-01

**Soundness:** 2
**Presentation:** 2
**Contribution:** 2
**Rating:** 2
**Confidence:** 4

**Summary:**

The paper proposes a way to build datasets and uses it to create an 8.5M-sample dataset for driving scenario, which focuses more on spatial relationships. Through experiments, the authors show that training on this dataset improves accuracy on the driving situation and also boosts performance on other external benchmarks that test spatial awareness.

**Strengths:**

This dataset is very large and seems to be high-quality. If the authors open-source it, it would be a great help to the community. It's also impressive that even though the data is only from the driving domain, it helps improve performance on general tasks.

**Weaknesses:**

1. The method in Algorithm 1 naively uses 2D bounding box alignment to infer "left/right" relationships, ignoring perspective. This is likely to introduce significant label noise in driving scenes by misinterpreting 3D "front/back" configurations as 2D "left/right" ones, leading to dataset inaccuracies.
2. I'm concerned about whether a detector like YOLO can actually tell apart different objects of the same type. For example, can it handle several cars that look almost identical? This must happen all the time in driving scenes. If the detector can't distinguish between them, how can the system create correct questions about their relationships? This seems like a major issue.
3. The reliance on an object detector for answer generation means that detection errors directly create factual inaccuracies (hallucinations) in the dataset. The paper lacks a quantitative evaluation of the answer generation accuracy.

**Questions:**

Please refer to the issues detailed in the Weakness part.

---

> ### Author Response · Authors · 2025-11-20
>
> We thank the reviewer for their feedback and recognizing both the scale and potential impact of GRAID.
>
> We address the numbered weaknesses in turn:
> ### (3) Reliance on object detectors and hallucinations / answer accuracy.
> The review states that “The paper lacks a quantitative evaluation of the answer generation accuracy.” Sections 4 and 5 of the existing submission are precisely aimed at quantifying this.
> - In Sec. 4, we conduct a human study where annotators judge both question validity and answer correctness for GRAID-BDD as well as for a recent spatial VQA pipeline. GRAID-BDD achieves over 91% valid QA pairs under a conservative evaluation protocol, whereas the prior pipeline achieves only 57.6% correct answers with 41.6% invalid questions.
> - In Sec. 5, we further validate answer quality indirectly by showing that training on GRAID systematically improves spatial reasoning and performance on multiple recent external benchmarks.
>
> Our reliance on 2D object detectors (and intentional avoidance of single-view 3D reconstruction) is precisely aimed at $\textit{reducing}$ hallucinations compared to existing pipelines. Prior work that depends on single-view 3D reconstruction or caption-based LLM generation compounds several sources of error while our design instead uses simple 2D geometry and avoids generative components in the data pipeline. As noted above, the human evaluation shows substantially fewer invalid / incorrect QA pairs for GRAID than for a recent 3D-based spatial VQA pipeline, and fine-tuning on our data leads to significant gains on established benchmarks. We will make this connection more explicit in the main text and clearly label these results as quantitative evaluation of answer correctness (see also Table 3 in the appendix).
>
> ### (2) Multiple similar objects of the same class.
> For our selected demonstration datasets (BDD100k, NuImages, Waymo), we use instance-level bounding boxes, where every object of a relevant class (e.g., each car) already has its own annotated box. Modern object detection benchmarks and methods are explicitly designed for this setting (see, e.g., the [survey by Zou et al., 2019](https://arxiv.org/abs/1905.05055)).
>
> ### ​​(1) 2D left/right vs. 3D perspective.
> In Algorithm 1, we chose to realize a template in which we ignore perspective and simply examine the 2D image plane and not 3D world coordinates. GRAID intentionally avoids single-image 3D reconstruction. It is the main reason we are able to produce datasets of significantly higher quality than currently published SOTA methods (as verified by our human study). Furthermore, the style of questions such as left or right matches the semantics of existing spatial VQA datasets such as BLINK. Note that, while we present a small set of generally applicable template based questions, our main contribution is the framework itself and not the templates. For example, in a domain specific setting in which you know the distribution of scenes (e.g. indoor living rooms), you can easily extend Algorithm 1 to accept a threshold where x1_min must be a certain margin larger than x2_max thus accounting for ambiguous settings (e.g. nearly collinear objects along a wall). We will clarify this in the paper and emphasize that our main contribution is the framework itself: users can easily propose templates with domain-specific predicates and realizers (e.g., tailored to indoor scenes or particular camera geometries).

---

### Official Review · Reviewer_Yb6x · 2025-11-02

**Soundness:** 3
**Presentation:** 1
**Contribution:** 3
**Rating:** 6
**Confidence:** 2

**Summary:**

This paper introduces GRAID, a data-generation framework that improves VLM spatial reasoning by operating on 2D detector boxes—avoiding single-view 3D reconstruction errors and caption-driven hallucinations. It scales via the proposed predicate-based engine (SPARQ) to create 8.5M+ VQA pairs across BDD100k, NuImages, and Waymo, with ~91% human-validated accuracy versus 57.6% for a recent pipeline. Fine-tuning on GRAID data yields some accuracy gains.

**Strengths:**

I believe improving the spatial reasoning abilities for VLM is important. I am surprised this pipeline described in this paper hasn't not been proposed (if true). Overall, I believe leveraging 2D models on this purpose technical sounds.

Their experimental results show that there are some cross-type transfer (e.g., training on 6 question types improves >10 held-out types), and also boosts public benchmarks such as BLINK and A-OKVQA.

Also, I appreciate the human validation results.

**Weaknesses:**

I was surprised to find that such a pipeline has not been studied before—or perhaps I am just not familiar with the relevant literature. I will double-check the related works and with other reviewers on this purpose.

The proposed template-based tasks are definitely limited the expression abilities for the datasets and the diversity.

Even 91% is not desirable in my mind for dataset quality, and particularly, the spatial reasoning tasks shown in the paper is not that challenging.

Extending the verified experiments beyond llama-3.2B 11B are helpful.

Also, the writing and figuring need to be improved.

**Questions:**

Plz address the concerns raised above.

---

> ### Author Response · Authors · 2025-11-20
>
> We thank the reviewer for their thoughtful and detailed comments, and for recognizing our experimental results in cross-type transfer and gains on public benchmarks. We also appreciate the reviewer engaging with work in a slightly different area.
>
> ## Novelty and relation to prior work
>
> We agree that several lines of work touch on spatial VQA. As summarized in Table 1, existing frameworks either (i) rely on single-view 3D reconstruction, or (ii) generate QA pairs from image–caption pairs via LLMs. Some prior works, such as SpatialVLM, do leverage 2D bounding boxes but only as one component in a more complex pipeline that ultimately targets $\textbf{quantitative}$, metric-based questions. In contrast, GRAID uses 2D boxes directly as the primary signal for generating $\textbf{qualitative}$ spatial questions. As a result, GRAID achieves much higher human-validated dataset quality than existing approaches. More importantly, this higher dataset quality leads to teaching spatial primitives with transferable skills, and improved scores in public benchmarks across a variety of VQA topics not just in spatial reasoning.
>
> Still, we will revise our writing to make our claim more precise: not that “2D spatial questions using bounding boxes has never been studied,” but rather that GRAID fills a current gap by proposing a higher-quality, simple-to-understand, template-based framework for large-scale spatial VQA generation from 2D detections. We will also take this opportunity to improve the clarity and design of our figures.
>
> ## Dataset quality
>
> As the reviewer noted, 91% is not perfect, however, achieving near perfect scores is impossible due to constraints outside of this submission. Whether we use ground truth labels or a well trained detector, we will not have perfectly annotated images as the datasets used to train detectors have annotation errors themselves.
>
> For example, [Northcutt et al., 2021](https://arxiv.org/abs/2103.14749), find a 5.83% error rate in the annotations of the ImageNet test set which was significant enough to cause issues in benchmarks and deployed models. Similarly, [Schubert et al., 2023](https://arxiv.org/abs/2303.06999), locate 34 labeling errors in BDD using just 200 proposals from their novel proposal method. In spite of such underlying noise, our method’s simplicity and avoidance of cascading 3D reconstruction errors allow us to achieve much higher data quality than prior spatial VQA pipelines. In our human study, the OpenSpaces dataset from SpatialVLM achieves only 57.6% correct answers with 41.6% invalid questions, whereas GRAID-BDD exceeds 91% valid QA pairs under a conservative evaluation protocol. Our study also agrees with concurrent findings (L368-372) that SpaceLLaVA, a VLM trained on SpatialVLM data, performs the worst among similarly sized models on spatial benchmarks.
>
> ## Simple Templates
>
> Although the surface forms of our questions are short, and the concepts are relatively simple, the true difficulty lies in answering these simple questions in real-world scenes which may be crowded and dense scenes (ex. Times Square NY with tens of objects present). Here a simple question requires identifying all relevant objects whether partially occluded or not, and then reasoning over their relative spatial relationships. As we report in our human study, (L386-389), the average perceived difficulty in answering these simple questions is not 1 or 2; nearly a third of the questions were rated of 4 or higher on a 5-point Likert difficulty scale.
>
> Moreover, as the reviewer noted, our initial template questions are deceptively simple yet still result in cross-type transfer and, more importantly, lead to improvements on external, challenging public benchmarks whose language and compositional structure go well beyond our templates. Our demonstrated improvements further validate that the learned concepts are genuinely useful and non-trivial despite the simple question forms.
>
> Lastly, we will rephrase our writing to emphasize that GRAID is a framework: our core library is easily extendable where users can use domain specific images, detectors, predicates, and realizers, to obtain more diverse and challenging QA pairs tailored to their own settings. Custom predicates and templates also offer users the opportunity to design templates robust to the types of errors they expect from their detectors.
>
> ## Additional Models
>
> We agree that evaluating on more than one VLM would strengthen the empirical results. Following your suggestion (and by others), we will extend our experiments to support an additional recent open-weight VLM and will report these results in the revised version. Since GRAID is model-agnostic, we expect the qualitative trends of improved spatial reasoning, and cross-type transfer from SFT on GRAID to hold across backbones.
>
> We hope these clarifications and planned additions address the reviewer’s concerns and better convey the novelty and practical impact of GRAID.

---

> > ### Comment · Reviewer_Yb6x · 2025-11-24
> > **Review response**
> >
> > I appreciate the authors’ response, but the concerns regarding data quality, the use of simple templates, and the evaluation on additional models were not addressed. Instead, the reply primarily emphasized future efforts, extensibility, and GRAID is a framework.
> >
> > I feel that, in its current form, the paper may not yet be mature enough for publication at this stage.

---

### Author Response · Authors · 2025-12-02

We thank all the reviewers for their thoughtful insights. A common theme across all reviews was that the paper would benefit from conducting our existing RQ3 experiment with an additional model to demonstrate that the results are not model specific. The other common insight was that, we would benefit from additional experiments to compare models against existing methods at the “capability-improvement level”. Lastly, their seemed to be lingering concerns about if models fine-tuned on spatial relationships from autonomous driving datasets, would in fact, understand spatial concepts in other indoor and outdoor scenes.

Thus, we extended our experiments and changed our submission in the following ways:
- We extend RQ3 to include a new additional benchmark:
  - [Visual Spatial Reasoning (VSR)](https://arxiv.org/abs/2205.00363): over 10,000 VQA pairs across both indoor and outdoor scenes.
  - We also adjust the paper to emphasize that the existing 4 benchmarks are not specific to autonomous driving.
- In addition to the prior selected Llama 3.2 11B model, we extend the experiments to include 3 new models:
  - Gemma 3 4B Instruction Tuned
  - Qwen 2.5 VL 3B Instruct
  - Qwen 3 VL 8B Instruct
- Capability-level comparison to prior spatial data
  - the base model
  - the model fine-tuned on OpenSpaces, a dataset generated by the community implementation of SpatialVLM
  - the model fine-tuned on GRAID-BDD
  - As noted in Table 1, the other two methods (SpaRE and SpatialRGPT) are either not open-source or require region-based prompting, so they cannot be used for a direct comparison.
- Scale of new experiments:
  - Across 5 benchmarks × (4 base models + 4 OpenSpaces-SFT models + 4 GRAID-SFT models) we now report 60 total experimental configurations. To keep the main paper readable, we summarize the key trends in the main text (RQ3) and provide the full results for all four backbones in Appendix Tables 4, 5, and 6.
- Findings and conclusions: The additional experiments reinforce our original claims.
  - Across all four backbones, GRAID-SFT models consistently outperform OpenSpaces-SFT models on most benchmarks and especially on spatial subtasks (e.g., BLINK Spatial Relation, Relative Depth, VSR). We emphasize that these benchmarks are not related to driving.
  - On stronger, newer models like Qwen 3 VL 8B, where spatial reasoning is already relatively strong, fine-tuning on GRAID yields little to no performance degradation, whereas fine-tuning on OpenSpaces often causes large regressions on both spatial and non-spatial tasks.
    - This directly addresses the request for capability-level comparisons: the GRAID-trained models improve or maintain performance where OpenSpaces-trained models frequently hurt or underperform GRAID models.

We also highlight that these gains come only from our initial set of 22 templates on driving images. Because GRAID is a framework, users can plug in their own domain-specific images, detectors, predicates, and realizers to construct richer template families and potentially obtain even larger improvements in their domain specific settings.

As Reviewer 8c6y noted, “*only demonstrating significant improvements in model capability can convincingly show the value of your data.*” We hope that the new experiments and explicitly comparing against the strongest publicly usable prior pipeline make it clear that models tuned on GRAID data consistently outperform models tuned on datasets generated by existing methods.

---

### Meta-Review · Area_Chair_xnc5 · 2025-12-28

**Summary:**

This submission received highly diverse scores (8642). The majors concerns raised by reviewers are about novelty of the approach, dataset quality and strength of empirical validation. After discussion, responses to major concerns from two reviewers were not responded. One reviewer raised new concerns about evaluation details and dataset quality while the other thinks that the limited scope of evaluation was not fully addressed.  After carefully reading the paper, the review comments and the discussions, I suggest the authors to carefully consider the concerns raised by reviewers and submit their work to the next conference.

**Reviewer Concerns:**

Some major concerns from reviewers were addressed by the rebuttal. One reviewer raised new concerns about evaluation details and dataset quality while the other thinks that the limited scope of evaluation was not fully addressed.

**Reviewer Scores:**

Two  reviewers responded to the rebuttal during the discussion phase, but there is no evidence to show the reviewers assigned negative scores would have changed their scores.

---

### Decision · Program_Chairs · 2026-01-26

Reject